# Comparative Study on Multiway Enhanced Bio- and Phytoremediation of Aged Petroleum-Contaminated Soil

**Natalia Ptaszek, Magdalena Pacwa-Płociniczak, Magdalena Noszczyńska and Tomasz Płociniczak \*** 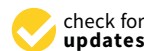

Faculty of Natural Sciences, Institute of Biology, Biotechnology and Environmental Protection, University of Silesia in Katowice, Jagiellońska 28, 40-032 Katowice, Poland; natalia-ptaszek@wp.pl (N.P.); magdalena.pacwa-plociniczak@us.edu.pl (M.P.-P.); magdalena.noszczynska@us.edu.pl (M.N.)

\* Correspondence: tomasz.plociniczak@us.edu.pl; Tel.: +48-322-009-442

**Abstract:** Bioremediation and phytoremediation of soil polluted with petroleum hydrocarbons (PHs) are an effective and eco-friendly alternative to physicochemical methods of soil decontamination. These techniques can be supported by the addition of effective strains and/or surface-active compounds. However, to obtain maximum efficacy of bioremediation, the interactions that occur between the microorganisms, enhancement factors and plants need to be studied. Our study aimed to investigate the removal of petroleum hydrocarbons from an aged and highly polluted soil (hydrocarbon content about 2.5%) using multiway enhanced bio- and phytoremediation. For this purpose, 10 enhanced experimental groups were compared to two untreated controls. Among the enhanced experimental groups, the bio- and phytoremediation processes were supported by the endophytic strain *Rhodococcus erythropolis* CDEL254. This bacterial strain has several plant growth-promoting traits and can degrade petroleum hydrocarbons and produce biosurfactants. Additionally, a rhamnolipid solution produced by *Pseudomonas aeruginosa* was used to support the total petroleum hydrocarbon loss from soil. After 112 days of incubation, the highest PH removal (31.1%) was observed in soil planted with ryegrass (*Lolium perenne* L. cv. Pearlgreen) treated with living cells of the CDEL254 strain and rhamnolipid solution. For non-planted experimental groups, the highest PH loss (26.1%) was detected for soil treated with heat-inactivated CDEL254 cells and a rhamnolipid solution. In general, the differences in the efficacy of the 10 experimental groups supported by plants, live/dead cells of the strain tested and rhamnolipid were not statistically significant. However, each of these groups was significantly more effective than the appropriate control groups. The PH loss in untreated (natural attenuation) and soils that underwent phytoremediation reached a value of 14.2% and 17.4%, respectively. Even though the CDEL254 strain colonized plant tissues and showed high survival in soil, its introduction did not significantly increase PH loss compared to systems treated with dead biomass. These results indicate that the development of effective biological techniques requires a customized approach to the polluted site and effective optimization of the methods used.

**Keywords:** bioremediation; bioaugmentation; phytoremediation; rhamnolipid; endophytes

## 1. Introduction

Petroleum is a mixture of various hydrocarbons and constitutes the principal raw material for chemical manufacturing and transportation in the global economy [1]. According to the U.S. Energy Information Administration, its worldwide production in 2019 was above 80 million barrels per day (www.eia.gov). Due to the extensive use of petroleum and processes related to its exploitation

(e.g., extraction, refining, transport, and distribution), petroleum hydrocarbons (PHs) have been recognized as one of the main sources of environmental pollution [2]. PHs are classified as organic contaminants with carcinogenic, mutagenic, and toxic effects [3].

Numerous attempts, including physical and thermal approaches, as well as chemical remediation methods, have been made to remediate PH-polluted soil. However, these procedures are costly, destructive to natural habitats and often lead to equally toxic secondary pollution [4,5]. As an alternative to the physicochemical techniques, the use of live organisms for remediation of polluted ecosystems, also known as bioremediation, is cost-effective and sustainable [6]. Several studies have successfully demonstrated the bioremediation of PH-contaminated soils through bioaugmentation, phytoremediation, or a combination of both [7–12].

Bioaugmentation involves the inoculation of exogenous hydrocarbon-degrading microorganisms to promote the removal of the pollutant from the environment [11]. However, the high hydrophobicity and low solubility of PHs result in their poor availability for bacterial cells, and thus, remediation of soil polluted with these compounds is limited [13]. An effective strategy that enhances contact surface between bacteria and PHs is the use of biosurfactants (BSs) [14]. BSs are amphiphilic molecules produced by microorganisms. Among them, the rhamnolipids (Rhs) synthesized by the *Pseudomonas* species can decrease the interfacial tension between the aqueous phase and PHs, and form a microemulsion whereby hydrocarbons are solubilized in water [15,16]. Since BSs are effective at bioremediation, many studies have used BS supplementation as a support for phytoremediation [17,18].

Phytoremediation is carried out by plants that can eliminate different contaminants, mainly by enhancing their rhizodegradation by microorganisms inhabiting the rhizosphere. The effective uptake of organic pollutants by plants is limited mostly due to their photoautotrophic metabolism and their inability to use organic molecules as a source of energy or carbon [19]. In our previous research, we have demonstrated that this limitation could be overcome using a combined plant–bacterial system [20]. The interactions between plants and bacteria both in the rhizo- and endo-sphere are important for the growth and productivity of plants, as well as for microbial remediation of PHs in the soil [1]. Due to the high amounts of various nutrients, enzymes, and other compounds secreted by plant roots, microorganisms are more likely to colonize the rhizosphere. Additionally, roots provide a greater surface area for bacterial growth and a sufficient quantity of oxygen [21]. Increase in the activity of the bacteria inhabiting the rhizosphere might result in enhanced biodegradation of some organic contaminants by a process referred to as rhizodegradation [22]. However, rhizodegradation often fails due to a low number of PH-degrading bacteria inhabiting such nutrient-rich environments. To overcome this problem, the inoculation of soil with PH-degrading and BS-producing bacteria is recommended. Additionally, these bacteria may act as plant growth-promoting bacteria (PGPB) [23]. Several PGPB not only colonize the root zone but can also enter plants and colonize their internal tissues. These bacteria enhance biodegradation of organic contaminants in the rhizosphere and/or endosphere, significantly accelerating the removal of organic pollutants from the soil [24]. Moreover, the rhizospheric and endophytic microorganisms, through their PGP activity, can compensate for the phytotoxicity of the contaminants [24,25].

Currently, the intensive development of phytoremediation methods of soils polluted with organic compounds is observed [26]. Other than the use of bioaugmented beneficial microorganisms to support these processes, a modern approach also involves the use of amendments such as a biochar [27,28], physical processes, like application of an electric field in electrokinetic-assisted phytoremediation [29] or the use of mixed surfactant solutions [30]. In all of these modified methods, ryegrass was also successfully used.

Microbial enhancement of bio- and phytoremediation has been shown to intensify the bioremediation strategy [9,31]. However, only a few studies have evaluated the effect of soil bioaugmentation with simultaneous BS application. In the present study, the effect of an endophytic PGP strain of *R. erythropolis* CDEL254 capable of PH degradation and BS production, together with Rh addition, on the efficacy of bioremediation of soil polluted with PHs was estimated. Ten experimental

groups of bio-and phytoremediation experiments supported with living cells of the tested strain (bioaugmentation), dead cells of the CDEL254 strain (biostimulation), and Rh solution (increase in PH bioavailability) with different combinations were tested to identify the most optimal strategy to support PHs removal from soil. This multiway treatment is a new approach in bioremediation of polluted soils; however, the use of such a technique requires an individual approach to the respective polluted areas.

## 2. Materials and Methods

### 2.1. Isolation, Biochemical Characterization, and Identification of a Hydrocarbon-Degrading Endophytic Strain

Hydrocarbon-degrading endophytic strains were isolated from the leaves of the *Lolium perenne*, growing near an oil refinery in Czechowice-Dziedzice, Upper Silesia, Southern Poland. After the surface sterilization of the plants as previously described by Kukla et al. (2014) [32], the macerated leaf extract was plated on M9 mineral salt medium supplemented with 1% (*v/v*) of crude oil as carbon and energy source [33]. Briefly, 20 mL of medium was introduced into a Petri dish containing 200 µl of crude oil. After mixing and solidifying of the medium, a thin layer of crude oil was formed on the medium surface. After seven days of incubation at 28 °C, the morphologically different bacterial colonies were reinoculated on M9 plates with 1% of crude oil.

The evaluation of the selected plant growth-promoting (PGP) traits of the endophytic strains was performed on selective media using the protocols previously described by Płociniczak et al. (2019) [34]. For endophytic isolates, the 1-aminocyclopropane-1-carboxylic acid deaminase (ACC) deaminase and cellulase activity were estimated. Additionally, siderophore secretion, indole-3-acetic acid (IAA) and ammonia production, phosphate solubilizing activity and motility of the isolates were examined. Briefly, ACC deaminase activity was assayed according to a modified Honma and Shinmomura method (1978) [35] as described by Saleh and Glick (2001) [36]. ACC deaminase activity was expressed in nmol of $\alpha$-ketobutyrate mg$^{-1}$ h$^{-1}$.The protein concentration of microbial cell suspensions was determined by the Bradford method (1976) [37]. The cellulase activity was examined by the inoculation of the isolates on carboxymethyl cellulose agar plates as described by Pointing (1999) [38]. Siderophore secretion by the tested strain was detected by the Schwyn and Neilands method (1987) [39] using blue agar plates containing the dye Chrome azurol S (CAS). Orange zones around the colonies on blue agar were considered a positive reaction for siderophore production. IAA production was determined according to the modified method of Bric et al. (1991) [40] using Salkowski's reagent. The IAA concentration in cultures was determined using a calibration curve of pure IAA (concentration range from 1 to 100 µg mL$^{-1}$ of medium) as the standard. Bacterial isolates were tested for the production of ammonia in peptone water according to Cappuccino and Sherman (1992) [41]. The phosphate solubilizing ability of the tested strains was determined on an NBRIP agar medium. Strains were stabbed using a sterile needle. The halo and colony diameters were measured after 14 days of incubation at 28 °C. Halo size was calculated by subtracting the colony diameter from the total diameter [42]. In order to determine the motility of the isolates, the bacteria were plated on nutrient medium plates containing agar at a 0.2% concentration. The diffusion of a colony was observed after 24 h [43].

Surface active properties were evaluated using the oil-spreading test and by determination of the emulsification capacity for *n*-hexadecane, diesel oil, and *p*-xylene (emulsification index E$_{24}$) according to the protocols described by Pacwa-Płociniczak et al. (2016) [33].

The ability of the isolates to reduce surface tension was evaluated using the Krűss K20 force tensiometer. Bacteria were incubated in M9 medium with 1% (*v/v*) crude oil on a rotary shaker at 120 rpm (28 °C) for five days. After incubation, the surface oil layer was removed, and the surface tension of the culture supernatant and non-inoculated M9 medium was measured.

Based on the results of the above tests, the most promising strain was chosen for pot experiments. CDEL254 was positive for all tested mechanisms, including degrading the crude oil in M9 medium with high efficiency and rapid multiplication under laboratory conditions. To classify the strain, 16S rRNA

sequence analysis was performed. Genomic DNA was extracted using GeneMatrix bacterial and yeast genomic purification kit (EURx). For 16S rRNA amplification, the universal bacterial primers 8F (5′ AGTTTGATCATCGCTCAG 3′) and 1492R (5′ GGTTACCTTGTTACGACTT 3′) [44] were used. The PCR was run with a mixture containing 1 μL of the DNA template, 0.2 μM of each primer, 10× reaction buffer (Thermo Scientific), 1.5 mM of $MgCl_2$ (Thermo Scientific), 200 μM of dNTP and 1 U of DreamTaq DNA polymerase (Thermo Scientific) in a C1000 Touch™Thermal Cycler (BioRad). PCR amplification was performed at 94 °C for 5 min, 3 cycles at 94 °C for 45 s, 57 °C for 30 s, 72 °C for 120 s; 3 cycles at 94 °C for 45 s, 56 °C for 30 s, 72 °C for 120 s; 3 cycles at 94 °C for 45 s, 55 °C for 30 s, 72 °C for 120 s; 26 cycles at 94 °C for 45 s, 53 °C for 30 s, 72 °C for 120 s; and a final elongation cycle at 72 °C for 5 min. Gene sequencing was performed by using the Big Dye® Terminator Cycle Sequencing Kit (Applied Biosystem) and the AbiPrism®3100 Genetic Analyzer. Next, the obtained sequence was compared with EZBioCloud database and a phylogenetic analysis was performed based on the CDEL254 sequence and nine closest type strains of other *Rhodococcus* species. The 16S rRNA sequences of the related type strains were obtained from GenBank. The analysis was performed using Mega X software using the maximum likelihood method, Tamura-3-parameter model, assuming that a certain fraction of sites are evolutionary invariable (+I) and 1000 bootstrap replicates [45,46]. The similarity of 16S rRNA sequences between the strains was calculated using the average nucleotide identity (ANI) calculator (www.ezbiocloud.net/tools/ani), [47].

## 2.2. Selection of a Rifampicin-Resistant Mutant of CDEL254 and Inoculum Preparation

A rifampicin-resistant mutant of the CDEL254 strain was selected to monitor the fate of the inoculant after soil bioaugmentation. In the preliminary studies we did not find rifampicin-resistant strains in soil and tissues of plants used in pot experiment. For this, the CDEL254 strain was inoculated on M9 mineral salt agar medium ($Na_2HPO_4$ 6 g, $KH_2PO_4$ 3 g, NaCl 0.5 g, $NH_4Cl$ 1 g, $MgSO_4 \times 7H_2O$ 0.24 g and $CaCl_2$ 0.01 g per liter of deionized water) supplemented with 10 μg mL$^{-1}$ of rifampicin and 1% (v/v) of crude oil. The colonies that grew were reinoculated on M9 plates with a progressively higher content of the antibiotic (up to 150 μg mL$^{-1}$). The stability of rifampicin-resistance was confirmed by sub-culturing the resistant strain CDEL254$^{rf}$ on M9 plates supplemented with crude oil in the absence of the antibiotic. CDEL254$^{rf}$ had the same biochemical features as the parental CDEL254. To simplify the description of the experiments, the rifampicin-resistant CDEL254$^{rf}$ strain has been henceforth labeled as CDEL254. In the next step, a bacterial inoculum was prepared for use in the pot experiments. CDEL254 was cultured on M9 medium supplemented with 150 μg ml$^{-1}$ of rifampicin and 1% (v/v) of crude oil in an orbital shaker at 120 rpm (28 °C) for 48 h. The number of bacteria in the inoculum was estimated based on optical density measurement and plating. An appropriate volume of the CDEL254 suspension was centrifuged (6000 rpm, 21 °C, 20 min, Sigma 4-16K), the bacterial pellet was washed twice with sterile distilled water and resuspended, depending on the treatment, in 25 or 50 mL of sterile water.

## 2.3. Experimental Setup

The bioremediation experiments were conducted using aged petroleum-contaminated soil (top layer, 0–20 cm depth) collected from an industrial area located around a 100 year old oil refinery in Czechowice-Dziedzice, Upper Silesia, Southern Poland. The soil was classified as silty clay loam and its detailed physicochemical characteristics have been described by Pacwa-Płociniczak et al. (2016) [33].

The bioremediation experiments including bioaugmentation, phytoremediation and natural attenuation variants with three biological replications were composed of the following treatments: (1) non-planted soil inoculated with the CDEL254 strain (bioaugmentation, B), (2) non-planted soil inoculated with heat-treated CDEL254 cells (control for bioaugmentation, B_C), (3) non-planted soil inoculated with the CDEL254 strain and rhamnolipid solution (bioaugmentation with rhamnolipid, BRh), (4) non-planted soil inoculated with heat-treated CDEL254 cells and rhamnolipid solution (control for bioaugmentation with rhamnolipid, BRh_C) and (5) non-planted soil treated with rhamnolipid

solution (SRh); (6) planted soil inoculated with the CDEL254 strain (enhanced phytoremediation, EP), (7) planted soil inoculated with heat-treated CDEL254 cells (control for enhanced phytoremediation, EP_C), (8) planted soil inoculated with the CDEL254 strain and rhamnolipid solution (enhanced phytoremediation with rhamnolipid, EPRh), (9) planted soil inoculated with heat-treated CDEL254 cells and rhamnolipid solution (control for enhanced phytoremediation with rhamnolipid, EPRh_C), (10) planted soil treated with rhamnolipid solution (PRh), (11) planted soil treated with sterile water (phytoremediation, P) and (12) non-planted soil treated with sterile water (natural attenuation, NA).

Before starting the pot experiment, the soil was air-dried and passed through a 2 mm sieve. Then the soil moisture was adjusted to 50% of the maximum water holding capacity and maintained at this level during the entire experiment. The pots (1 L of volume, 11 cm high, 9.5 and 12.5 cm of bottom and top diameter, respectively) were filled with 500 g (d.w.) of polluted soil (about 800 cm$^3$). Seeds of *Lolium perenne* L. cv. Pearlgreen (35 per pot), taken from Lesser Poland Plant Breeding LLC, were placed into the soil and then covered with a 1-cm layer of soil. The experiment was conducted in a growth room under controlled light (14 h photoperiod at 15,000 lux; temperature 23/18 °C; light/dark). After 2 weeks, the number of plants was standardized to 30 per pot, and depending on the experimental group, (i) 50 mL of the bacterial solution of CDEL254 ($10^7$ cfu g$^{-1}$ d.w. of soil), (ii) mixture of 25 mL two-fold concentrated bacterial suspension ($10^7$ cfu g$^{-1}$ d.w. of soil) and 25 mL two-fold concentrated rhamnolipid solution (5 µg g$^{-1}$ d.w. of soil), (iii) 50 mL of rhamnolipid solution (5 µg g$^{-1}$ d.w. of soil) or (iv) 50 mL of sterile water was added to the soil. The experiment was carried out for 112 days. Soil and plant (roots and shoots) samples collected on days 1, 10, 30, 60, 90 and 112 were immediately acquired for estimation of the survival of strain in the soil and for determination of their ability to colonize the tissues of ryegrass. In the case of perennial ryegrass, the aboveground parts of the plant named here as shoots are mainly leaves Additionally, whole plants, divided into root and shoot subsamples, were collected on day 112 for the determination of plant biomass. Soil samples from days 0 and 112 were collected for the measurement of hydrocarbon concentration and from days 1 and 112 were used for quantification of the 16S rRNA copy number. The soil samples were stored at −80 °C.

### 2.4. Effect of the Applied Treatment on Plant Biomass

At the end of the experiment, the plants were removed from the pots and washed with sterile water. The shoots and roots were separated, and the wet and dry weight of these parts were determined.

### 2.5. Effect of the Bioremediation on the Removal of Petroleum Hydrocarbons

The total petroleum hydrocarbon concentration in the soil was measured for each pot before and after bioremediation. Hydrocarbons with a carbon number between 10 and 40 (TPH$_{C10-C40}$) were estimated following the ISO 16703:2011 according to Płociniczak et al. (2017) [20].

### 2.6. Survival of CDEL254 in the Soil and its Ability to Colonize Plant Tissues

The number of living CDEL254 cells was determined in the soil, roots, and shoots of the ryegrass on days 1, 10, 30, 60, 90 and 112 after inoculation according to Płociniczak et al. (2019) [34].

### 2.7. The Impact of Bioremediation on the Total Bacterial Load in the Soil

Real time PCR was used to analyze changes in the total number of bacteria in the soil on days 1 and 112 of the experiment. For this purpose, DNA was extracted from experimental soils, in three replicates, using a DNeasy PowerSoil Kit (Qiagen, Germantown, MD, USA) according to the manufacturer's instruction. The yield and purity of the DNA were determined using a NanoDrop ND-1000 spectrophotometer (NanoDrop Technologies, Wilmington, DE, USA). Quantification of the 16S rRNA copy number was performed using specific primers pE (5′-AAA CTC AAA GGA ATT GAC GG-3′) and pF (5′-ACG AGC TGA CGA CAG CCA TG-3′) [48] as described by Pacwa-Płociniczak et al. (2016) [49].

*2.8. Statistical Analysis*

Statistical analysis was performed using the STATISTICA 13.1 PL software (StatSoft, Tulsa, OK, USA). Analysis of variance (ANOVA) followed by a post-hoc least significant difference test was conducted to identify any significant effects of the individual treatments on the plant biomass, TPH removal, and the number of 16S rRNA genes. Differences between the control groups (P and NA) and those from the remaining experiments with a $p < 0.05$ were considered significant. For the pot experiments, data are represented as the mean ± standard deviation (SD) of three biological replicates.

## 3. Results

*3.1. Identification and Biochemical Characterization of CDEL254*

Phylogenetic analysis showed that the 16S rRNA sequence of CDEL254 (MT020429) had 99.5% identity with *Rhodococcus degradans* CCM4445 and *Rhodococcus qingshengii* JCM15477 and 99.3% identity with *Rhodococcus erythropolis* JCM3201 (Figure 1). However, the results of selected biochemical tests using the Biolog® system and API testing, as well as the fatty acid composition analysis, strongly support the identification of the CDEL254 strain as *Rhodococcus erythropolis*.

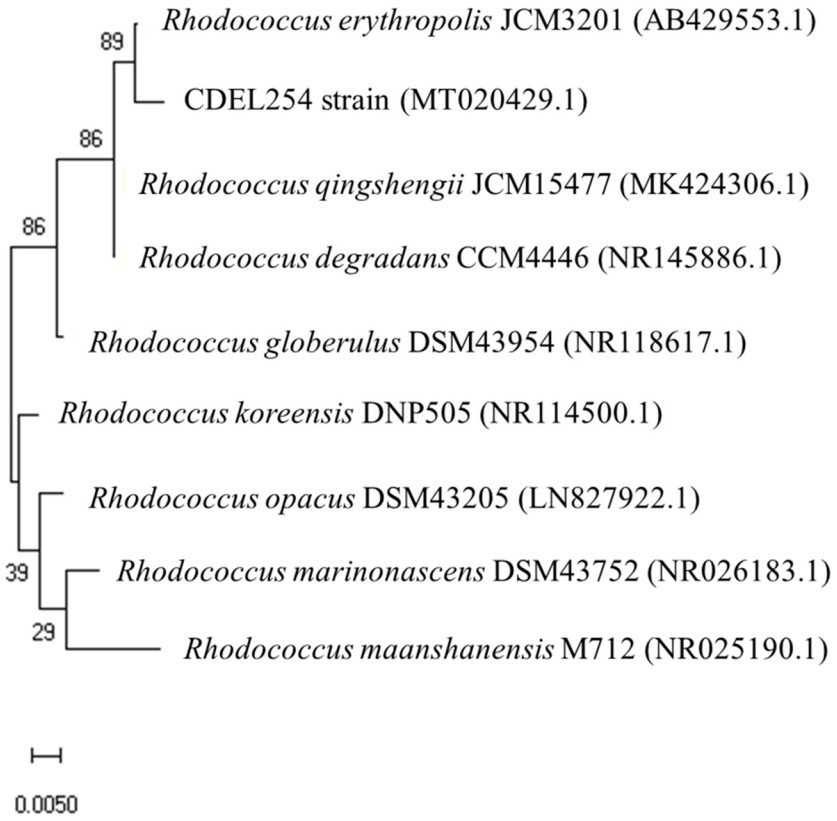

**Figure 1.** Phylogeny of strains closely related to *Rhodococcus erythropolis* CDEL254 based on 16S rRNA sequences. All positions containing gaps or missing data were removed. The length of the final dataset was 1238 bp. Bootstrap values are presented at the branch points. The bar represents 0.0050 substitutions per site, accession numbers from GenBank are shown in brackets.

Biochemical characterization of CDEL254 showed very high ACC deaminase activity (242.0 ± 12 nmol $\alpha$-ketobutyrate mg$^{-1}$ h$^{-1}$) and demonstrated the ability of this endophytic strain to produce indole 3-acetic acid (32.0 ± 3.7 µg IAA ml$^{-1}$ of medium), hydrocyanic acid, ammonia, and cellulase. Additionally, the motile CDEL254 solubilized Ca$_3$(PO$_4$)$_2$ (the diameter of the clear zone around the colony was 21.3 ± 2.1 mm), emulsified the tested hydrocarbons (*n*-hexadecane, 18% ± 1.5%,

diesel oil, 28.5% ± 2.3%, and *p*-xylene, 54.2% ± 1.8%) and generated a clear zone with a diameter of 23.5 ± 1.5 mm in the oil-spreading test. The CDEL254 strain reduced the surface tension of M9 medium containing crude oil to 54.8 ± 0.62 mN m$^{-1}$, while the non-inoculated control medium had a surface tension of 70.5 ± 0.1 mN m$^{-1}$.

### 3.2. The Impact of the Applied Treatments on Plant Biomass

Relative to the *L. perenne* planted soil (P), addition of bacteria, both living and heat-treated (EP, EP_C), significantly ($p < 0.05$) increased the fresh weight of the shoots by 27% and 10%, respectively, while the addition of rhamnolipid as the single factor (PRh) and together with CDEL254 strain (EPRh) significantly reduced fresh weight of the shoots by 7.5%, compared to P (Figure 2). The dry weight of the shoots was significantly ($p < 0.05$) enhanced only when live CDEL254 cells were added, compared to other treatments between which no significant differences were found. The fresh and dry weight of the roots was significantly enhanced, compared to planted soil (P), when both living and heat-treated (EP, EP_C) bacteria were added. However, the fresh weight of the roots was significantly reduced, compared to P treatment when rhamnolipid was introduced (PRh), but no significant reduction in the dry weight of the roots between P soil and EPRh, as well as EPRh_C treatments was observed.

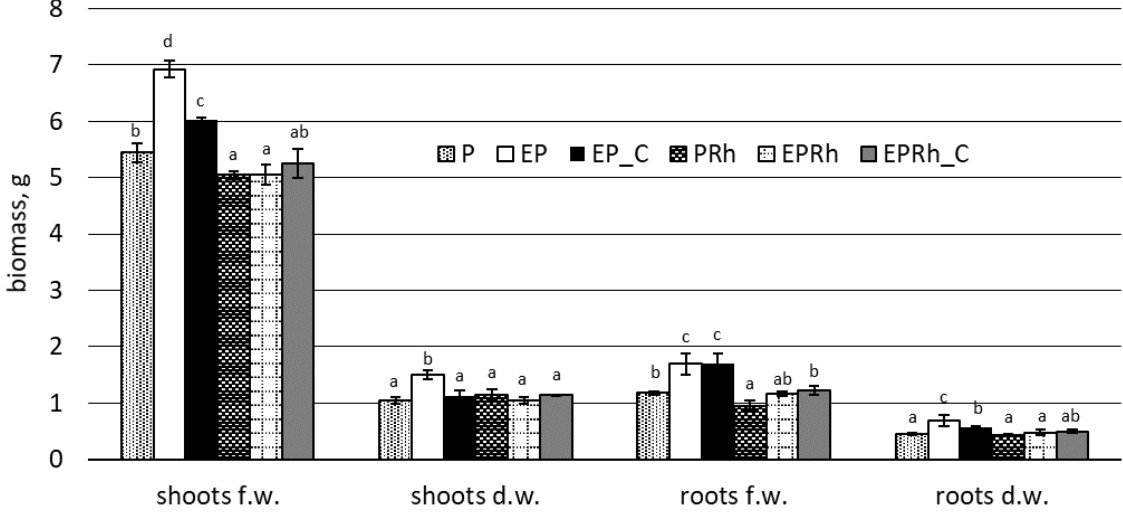

**Figure 2.** Biomass of shoots and roots of ryegrass that were grown in treated and control (P) soil. P, phytoremediation; EP, phytoremediation enhanced by live CDEL254 cells; EP_C, phytoremediation enhanced by dead CDEL254 cells; Rh, rhamnolipid; (mean ± SD, *n* = 3); f.w., fresh weight; d.w., dry weight. Different letters (within each group) indicate significant differences ($p < 0.05$, LSD test).

### 3.3. The Impact of the Applied Treatment on TPH Removal

The soil subjected to the bioremediation processes had a high initial TPH concentration (25758.05 ± 1993.34 mg kg$^{-1}$ d.w. of soil). The effect of different treatments on TPH degradation after 112 days is shown in Figure 3. The highest efficiency of TPH removal (31.1% ± 1.1%) was seen for planted soil inoculated with the CDEL254 strain and rhamnolipid (EPRh). Nevertheless, nonsignificant ($p < 0.05$) differences in the efficiency of TPH removal were observed between EPRh and B, BRh, BRh_C, SRh, EP, EP_C, EPRh_C and PRh, where removal of hydrocarbons was within the range 22.9–29.5%. Significantly ($p < 0.05$) lower TPH removal, compared to the above treatments was reported for soils NA, P and B_C and reached values of 14.2% ± 1.9%, 17.4% ± 6.1% and 16.3% ± 8.7%, respectively.

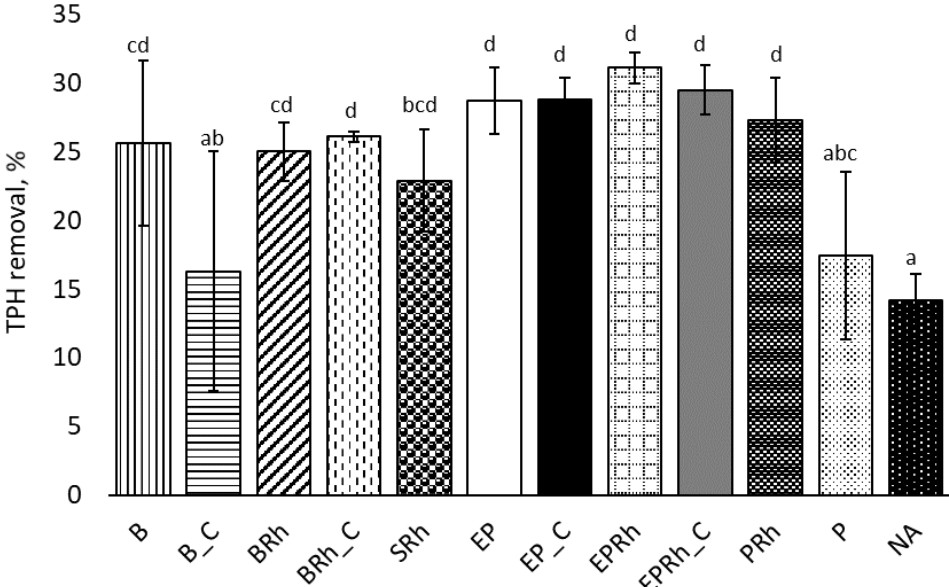

**Figure 3.** TPH removal efficiency in bioremediation treatments. B, bioaugmentation with live CDEL254 cells; B_C, biostimulation with dead CDEL254 cells; S, soil; P, phytoremediation; EP, phytoremediation enhanced by live CDEL254 cells; EP_C, phytoremediation enhanced by dead CDEL254 cells; Rh, rhamnolipid; NA, natural attenuation; (mean ± SD, *n* = 3). Different letters (within each group) indicate significant differences (*p* < 0.05, LSD test).

*3.4. Survival of CDEL254 in the Soil and the Colonization of Ryegrass Tissues*

The rifampicin-resistant mutant of the CDEL254 strain was tested for its ability to survive in soil and to colonize roots and shoots of *L. perenne* (Figure 4 A,B). The strain survived in the soil during the entire experimental period for EP and EPRh treatments, while for B and BRh treatments, the CDEL254 strain was isolated on medium supplemented with rifampicin up to day 90 and 60 of the experiment, respectively (Figure 4A). Nevertheless, regardless of the treatment, the gradual decrease in CDEL254 cell number during the experimental period was observed. The day after initial soil inoculation, the number of cells decreased by about one order of magnitude for EP, EPRh, and B and reached values of 6.13, 6.17, and 6.01 $\log_{10}$ cfu $g^{-1}$ d.w. of soil, respectively, and nearly two orders of magnitude for BRh treatment (5.53 $\log_{10}$ cfu $g^{-1}$ d.w. of soil). At the next sampling points (day 10 and 30), the number of CDEL254 cells calculated in the soil from EP, EPRh, and B treatments decreased slightly and remained at about 5.68, 5.86, and 5.4 $\log_{10}$ cfu $g^{-1}$ d.w. of soil, respectively. At the same time, the number of CDEL254 cells in soil from BRh treatment was two orders of magnitude lower and reached a value of 3.6 $\log_{10}$ cfu $g^{-1}$ d.w. of soil. A count of the inoculant in the soil from EP and EPRh treatments on days 60, 90, and 112 revealed that cell number remained at 4.42–4.82 $\log_{10}$ cfu $g^{-1}$ d.w. of soil. In contrast, the number of CDEL254 cells calculated in soil B decreased on days 60 and 90 and after which the strain was not detected in B, while its number in soil BRh was estimated to be very low (1.38 $\log_{10}$ cfu $g^{-1}$ d.w. of soil) on day 60 and thereafter the strain could not be isolated on medium supplemented with rifampicin.

*R. erythropolis* CDEL254 strain was able to colonize the roots and shoots of *L. perenne*; however, its colonizing ability differed depending on the treatment (Figure 4B). Although CDEL254 cells were detected in the roots from both EP and EPRh treatments, in the case of EP, they were detected up to day 60 and in the case of EPRh, CDEL254 cells were present in the roots of the *L. perenne* during the entire experimental period. The number of CDEL254 cells calculated in the roots of ryegrass one day after soil inoculation was estimated at 2.20 and 2.78 $\log_{10}$ cfu $g^{-1}$ d.w. of root for EP and EPRh, respectively. Then, its number increased and reached 3.44 and 3.98; 4.14 and 4.43; 3.53 and 4.19 $\log_{10}$ cfu $g^{-1}$ d.w. of root for EP and EPRh on days 10, 30, and 60, respectively. On day 90 and 112, bacterial cells were

only detected in the roots of ryegrass form the EPRh treatment group and reached values of 4.00 and 3.54 $\log_{10}$ cfu g$^{-1}$ d.w. of root, respectively.

(**A**)

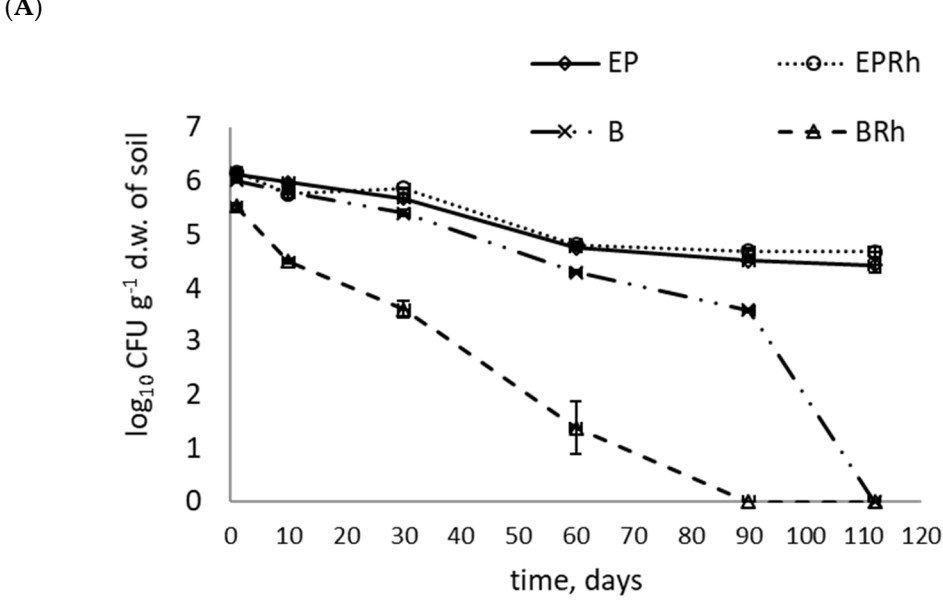

(**B**)

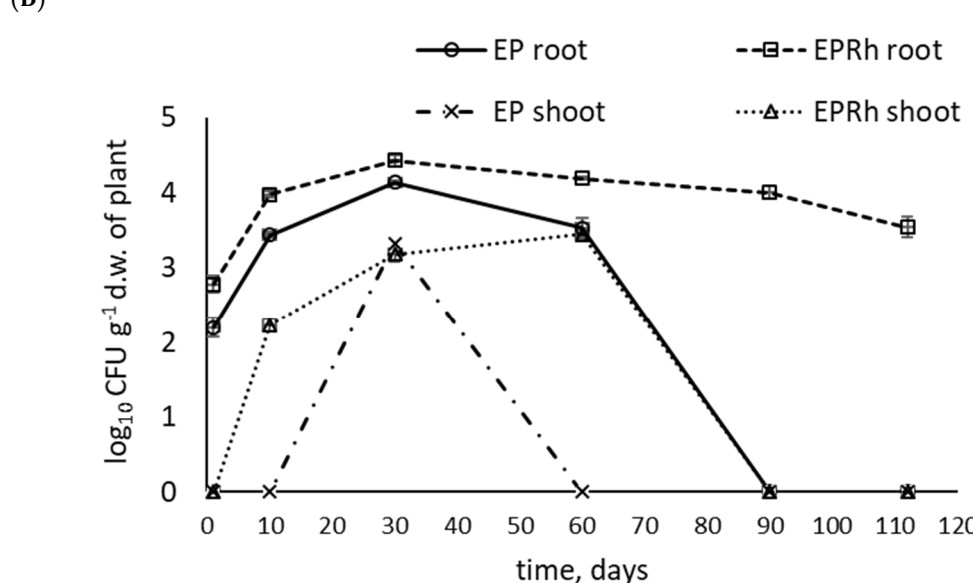

**Figure 4.** Number of rifampicin-resistant CDEL254 cells isolated from the soil (**A**) and plant tissues (**B**) during the bioremediation experiments. EP, phytoremediation enhanced by living CDEL254 cells; B, bioaugmentation with living CDEL254 cells; Rh, rhamnolipid; (mean ± SD, *n* = 3).

In the case of the shoots of the *L. perenne*, no rifampicin-resistant bacteria were isolated on day one after soil treatment with CDEL254. Ten days after soil inoculation, cells of *R. erythropolis* CDEL254 were only detected in the shoots of ryegrass from EPRh (2.23 $\log_{10}$ cfu g$^{-1}$ d.w. of shoot). On day 30, cells of the inoculant were present in the shoots of plants from both EP and EPRh treatments and reached values of 3.33 and 3.18 $\log_{10}$ cfu g$^{-1}$ d.w. of shoots, respectively. On day 60, similar to day 10, rifampicin-resistant bacteria were only isolated from shoots of ryegrass with EPRh treatment (3.45 $\log_{10}$ cfu g$^{-1}$ d.w. of shoot). After that, no CDEL254 was detected in the shoots of ryegrass from both treatments.

### 3.5. The Impact of the Applied Treatment on the Total Bacterial Number in the Soil

To estimate the impact of the applied bioremediation treatment on the total number of bacteria in the soil, the number of copies of the 16S rRNA gene g$^{-1}$ d.w. of soil was quantified (Figure 5A,B). On day one, content of the 16S rRNA gene was the highest in the non-planted soil inoculated with heat-treated CDEL254 cells (B_C), planted soil treated with sterile water (P), and planted soil inoculated with heat-treated CDEL254 cells (EP_C), and reached a value of 10.70 log$_{10}$ gene copies g$^{-1}$ d.w. of soil (Figure 5A). The amount of 16S rRNA genes estimated in the above soils was significantly ($p < 0.05$) higher compared to the non-planted soil treated with sterile water (NA) (10.65 log$_{10}$ gene copies g$^{-1}$ d.w. of soil). Significantly lower gene numbers were observed in non-planted soil inoculated with both live and heat-treated CDEL254 strains following addition of rhamnolipid (BRh and BRh_C) (10.54 and 10.45 log$_{10}$ gene copies g$^{-1}$ d.w. of soil, respectively) and in non-planted soil treated with rhamnolipid (SRh) (10.40 log$_{10}$ gene copies g$^{-1}$ d.w. of soil), compared to soil NA. Finally, no significant ($p < 0.05$) differences were observed on day one between soil NA and soils B, EP, EPRh, EPRh_C, PRh.

Quantification of the 16S rRNA copy number in the samples from day 112 showed that the soils B_C and EP_C (on the level of 10.72 log$_{10}$ gene copies g$^{-1}$ d.w. of soil) still had the highest content of this gene which was significantly higher than in soil subjected to natural attenuation (10.68 log$_{10}$ gene copies g$^{-1}$ d.w. of soil) (Figure 5B). Similar to day one, nonsignificant differences were observed on day 112 between soil NA and soils B, EP, EPRh and between soils NA and P. Moreover, just as on day one, statistically lower 16S rRNA copy number was observed in soils BRh, BRh_C, and SRh (10.46, 10.55, 10.41 log$_{10}$ gene copies g$^{-1}$ d.w. of soil, respectively), and in soils EPRh_C and PRh (10.60 and 10.57 log$_{10}$ gene copies g$^{-1}$ d.w. of soil, respectively) compared to soil NA.

(**A**)

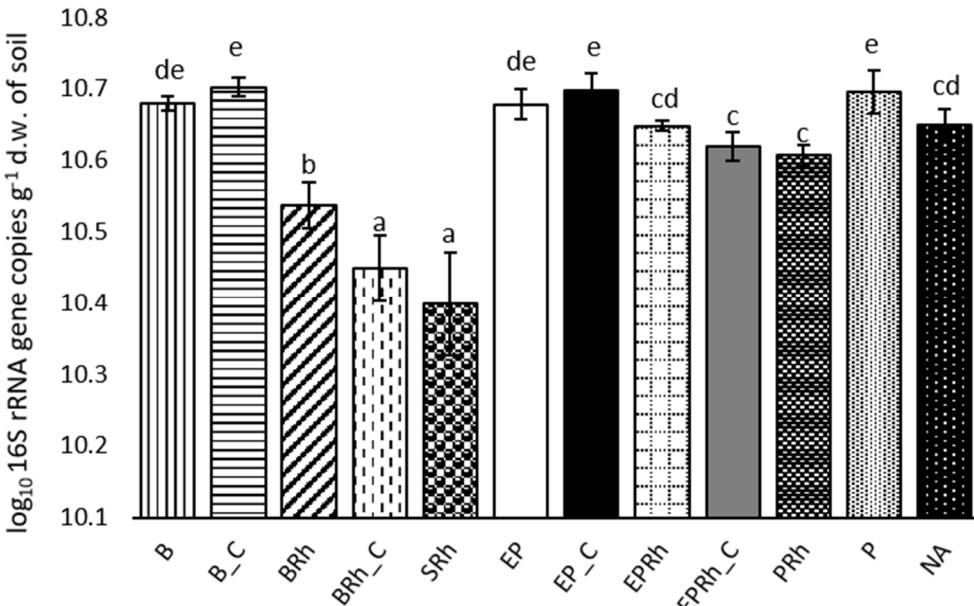

**Figure 5.** *Cont.*

(**B**)

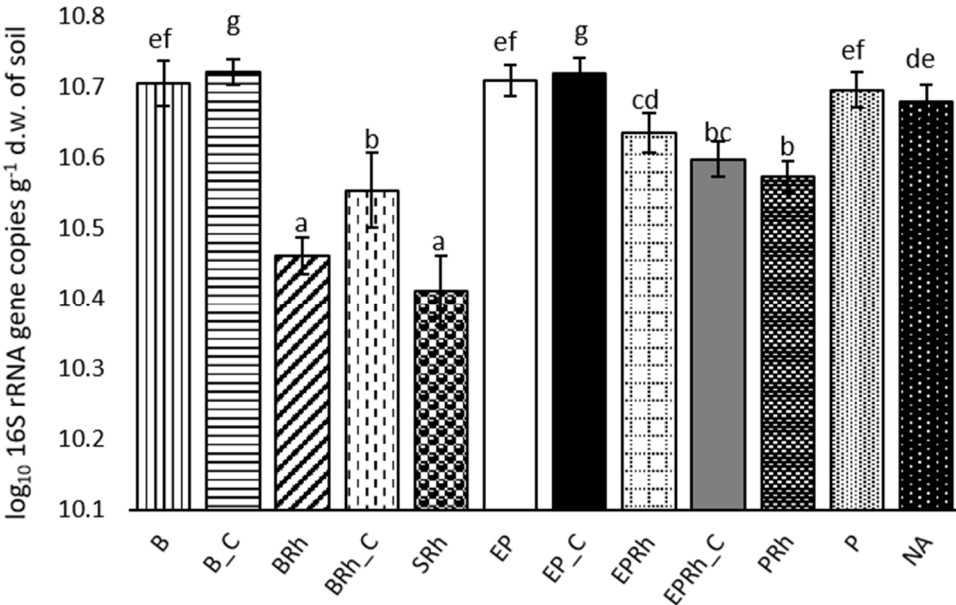

**Figure 5.** Number of 16S rRNA copies in soil 24 h after treatment (**A**) and after 112 days of incubation (**B**). B, bioaugmentation with living cells of the CDEL254 strain; B_C, biostimulation with dead cells of the CDEL254 strain; S, soil; P, phytoremediation; EP, phytoremediation enhanced by living cells of the CDEL254 strain; EP_C, phytoremediation enhanced by dead cells of the CDEL254 strain; Rh, rhamnolipid; NA, natural attenuation; (mean ± SD, $n$ = 3). Different letters (within each group) indicate significant differences ($p < 0.05$, LSD test).

## 4. Discussion

The results of several studies on the treatment of contaminated environments indicated that biological techniques, including phytoremediation and biodegradation, are suitable to clean up soil polluted with organic compounds. However, they can be effective only in soils with limited contamination and under no time constraints. To accelerate the rate of environmental restoration, these methods can be enhanced by introducing microorganisms exhibiting useful traits, such as the ability to degrade hydrocarbons, production of BSs, and in the case of phytoremediation, plant growth-promotion. Such multidimensional support for the bioremediation methods increases the efficiency of these processes and reduces their duration.

In the present study, the effect of the endophytic *Rhodococcus erythropolis* CDEL254 on the efficiency of several bioremediation processes was tested. This strain was able to degrade the PHs and produce surface-active compounds. Additionally, the CDEL254 strain exhibited several plant growth-promoting activities. Bacteria from the *Rhodococcus* genus are recommended for enhancing bioremediation of hydrocarbon-polluted soils since they are the predominant microorganisms in many polluted ecosystems and synthesize nonionic glycolipid complexes containing trehalose corynomycolates as major surface-active compounds. Moreover, they are safe to use (*R. erythropolis* belongs to risk group one), whereas the effective rhamnolipid producer *Pseudomonas aeruginosa* belongs to risk group two and can act as an opportunistic human pathogen [50]. Additionally, as confirmed in previous studies, strains of the *Rhodococcus erythropolis* species can exhibit several PGP mechanisms, even if they are isolated from bulk soil polluted with PHs [33].

It has been reported that the addition of BSs, such as rhamnolipids, may increase the uptake of hydrocarbons from soils by bacteria and/or plants by facilitating the mass transfer of these compounds from the solid into the aqueous phase [51] Therefore, in the present experiment, in addition to

the *R. erythropolis* CDEL254 strain, the rhamnolipid was tested as an enhancing agent in bio- and phytoremediation of PH-polluted soil.

### 4.1. The Impact of the Applied Treatment on the Plant Biomass and TPH Removal

The promoting effect of *R. erythropolis* CDEL254 strain on fresh biomass of ryegrass was evident by the significant increase in the weight of the shoots of plants growing in soil treated with living (EP) and heat-treated (EP_C) CDEL254 cells which was 27.2% and 10.5% higher compared to the untreated control (P), respectively. The same tendency, but a more pronounced effect was also observed for fresh biomass of the roots. The biomass of the roots from EP and EP_C experimental groups was 43.6% and 40.2% higher than in control plants. A different effect was observed in the case of plants supplemented with rhamnolipid. The fresh biomass of the shoots of plants cultivated in soils treated with rhamnolipid (PRh and EPRh) was about 8% lower compared to the control. An even higher decrease (19%) was detected for the biomass of the roots of plants growing in soil PRh. The impact of bioaugmentation on plant biomass observed in EP variant was statistically significant compared to the control, but it was smaller than the result obtained by Tang et al. (2010) [10]. In that study, the biomass of ryegrass treated with the *R. erythropolis* strain after day 162 of the experiment was two-fold higher than the control. Moreover, the biomass of plants treated with the consortium of four effective strains (composed of *Bacillus subtilis*, *Sphingobacterium multivorum*, *Acinetobacter radioresistens*, and *Rhodococcus erythropolis* strains) was more than three-fold higher as compared to non-inoculated control plants. The results obtained by Zhen et al. (2019) [52] showed the positive effect of biochar and rhamnolipid on the biomass of *Spartina anglica* after day 60. The plants were cultivated in soils containing 0, 10, 30, and 50 g TPH kg$^{-1}$ and the enhancement of shoot and root length was observed in soils treated with low and medium concentrations of PHs. They found that the addition of biochar, a mixture of biochar and rhamnolipid, and rhamnolipid modified biochar into medium-polluted soil (30 g TPH kg$^{-1}$) increased the shoot length of the treated plants by 35.7%, 46.5%, and 59.7%, respectively. However, based on these studies, it is difficult to clearly assess the impact of rhamnolipid alone on plant biomass. A negative effect of rhamnolipid on plant development was observed by Liduino et al. (2018) [31]. In their experiment with sunflower and hydrocarbon-polluted soil treated with rhamnolipid at 4 mg kg$^{-1}$, germination of seeds was not detected. For this reason, plants were germinated in garden soil and the seedlings were then transplanted to the polluted soil treated with rhamnolipid. After 90 days they found that the rhamnolipid, at the mentioned concentration, still had a negative influence on the growth of sunflower plants, probably due to its direct toxicity to microorganisms and plants or due to rapid mobilization of high amounts of organic pollutants.

In general, most of the bioremediation (except for B_C) and all phytoremediation experiments were significantly more effective at removing PHs from the soil than their control non-treated groups (NA and P, respectively). Although the differences in the hydrocarbon loss between individual experimental groups (except for B_C) were not statistically significant, a larger percentage loss of PHs was observed for phytoremediation systems and the highest PH removal (31.1%) was observed for plants treated with living CDEL254 cells with simultaneous Rh addition. Among non-planted experimental groups, the highest PH removal (26.1%) was observed for soil treated with dead biomass of CDEL254 together with rhamnolipid. The higher efficiency of bioaugmentation and phytoremediation processes over natural attenuation was also reported by Rodriguez-Campos et al. (2019) [53] for soil with low TPH concentration (309 mg kg$^{-1}$). After 112 days of incubation, the TPH loss for non-treated polluted soil reached a value of 27.1%, whereas for the remaining bioremediation experimental groups this value was significantly higher and reached more than 72%. The highest, 87% and 86.7% loss of the initial TPH concentration at the end of the experiment were observed for sterile soil treated with a consortium of four petroleum-degrading strains (C6, C9, C28, and C39) and for sterile soil planted with *Panicum maximum* and inoculated with above-mentioned strains, respectively. The effectiveness of these variants was significantly higher compared to other experimental sets described in this study and showed no significant influence of the plant on the loss of PH from soil contaminated with small amounts of

organic pollutants. Both bioaugmentation and phytoremediation enhanced the degradation rate of TPH compared to untreated controls in bioremediation experiments performed by Tang et al. (2010) [10]. They tested several bacterial strains and their mixtures to support the bioremediation processes of soil with high total petroleum content (6.19%). In experiments with *Rhodococcus erythropolis*, after 162 days of incubation, the TPH removal for bacterial-assisted phytoremediation and bioaugmentation reached a value of about 55% and 52%, respectively, indicating a small effect of plants on bioremediation efficacy. For comparison, TPH removal from non-supported phytoremediation and natural-attenuated groups reached a value of about 48% and 41%, respectively. The same tendency was observed in the similar experimental group enhanced by *Bacillus subtilis*; however, when the consortium of *Sphingobacterium multivorum* and *Acinetobacter radioresistens* was used as the inoculum, the highest TPH loss was observed.

The bioremediation of soils polluted with PHs can also be enhanced by surface-active compounds produced by microorganisms after inoculation or implemented as a BS solution prepared earlier. In the case of the phytoremediation experiment performed in our study, the treatment of the soil with the rhamnolipid solution (PRh) caused a significant increase in the effectiveness of TPH removal from the soil, compared to control (P), probably by increasing the availability of pollutants for autochthonic microorganisms and plants. Slightly higher TPH loss was observed for the remaining rhamnolipid-treated phytoremediation experiments (EPRh, EPRh_C) compared to variants without its addition (EP, EP_C). Similarly, for non-planted soil, the TPH loss in soils treated with rhamnolipid (BRh, BRh_C, SRh) was significantly higher compared to control (NA) and for multiway enhanced groups the rhamnolipid use in BRh_C caused significantly higher TPH removal compared to B_C.

A crude BS extract produced by hydrocarbon-degrading bacteria *Serratia marcescens* was tested as an additive during the phytoremediation of PHs using *Ludwigia octovalvis* by Almansoory et al., (2019) [54]. After 72 days of incubation, the PH loss in soil treated with the undefined BS produced by *S. marcescens* and sodium dodecyl sulfate (SDS) reached a value of 93.5% and 86.2%, respectively. The same authors previously tested the impact of (1) bioaugmentation with *S. marcescens* strain, as well as (2) the supernatant after bacterial cultivation, (3) pure BS solution, and (4) SDS solution on the effectiveness of phytoremediation in soil spiked with 0.2% of gasoline [55]. They reported that the highest (93.5%) loss of PHs at the end of the experiment was observed for planted soil treated with a BS solution. The PH removal from soil treated with SDS and *S. marcescens* strain was around 85% and was higher compared to the addition of the bacterial supernatant.

The positive effect of two BSs (rhamnolipid and soybean lecithin) on the phytoremediation efficacy of soil polluted with crude oil at a concentration of 5000 mg kg$^{-1}$ was also reported by Liao et al. (2016) [56]. After three months of incubation of *Zea mays* plants in polluted soils treated with surface active compounds, the PH removal efficacy in the treatment with soybean lecithin, rhamnolipid, and Tween 80 were 62%, 58%, and 47%, respectively. At the same time, the TPH loss in non-supported phytoremediation was 52%, while the efficacy of natural attenuation of non-planted soil was significantly lower and reached a value of 13%. The use of BSs did not cause the higher PAH accumulation in maize leaf. Moreover, their concentration was lower than in control plants, which suggests that the PAH loss from the soil was not caused by a higher uptake of pollutants by the plants, but rather by their better availability and higher degradation by rhizospheric microorganisms.

*4.2. Survival of CDEL254 in the Soil and the Colonization of Ryegrass Tissues*

There is limited information available on whether a strain used for microbial bioremediation experiments can survive in the soil, enter the plant and then colonize its tissues. Generally, the number of introduced bacteria sharply declines after the initial soil inoculation. This decrease in colony forming units (CFU) is not well understood. Recent studies by Fida et al. (2017) [57] using genome-wide transcription analysis and direct counting of *Novosphingobium* sp. LH128 tagged with GFP confirmed that a decline in CFU numbers represents a transition of LH128-GFP into a viable but nonculturable (VBNC)-like state. Many bacteria enter the VBNC state to cope with adverse environmental conditions

but can still exert their metabolic activity [58]. In the present study, the fate of the CDEL254 strain after soil inoculation was tracked by plating soil and plant extracts on selective medium. Using this technique, we were not able to estimate how many cells entered the VBNC state, but this method is still useful in estimating the potential of CDEL254 to survive in soil and colonize the roots and shoots of ryegrass. Live CDEL254 cells ($10^7$ cfu g$^{-1}$ d.w. of soil) were used in the EP, EPRh, B, and BRh experiments. At the first sampling point, a decrease in the number of CDEL254 cells by one order of magnitude was observed in the soil for EP, EPRh, and B experiments. An even higher decrease was observed for the BRh group. For groups in which plants were used, live CDEL254 could be isolated from the soil even on the last day of the experiment, whereas in non-planted experiments its survival in soil was lower. This phenomenon might be due to more favorable conditions present in the rhizosphere, which creates a niche for the development of microorganisms. The roots of plants exude many easily degradable carbon and energy sources, which can support bacterial growth. In such an environment, bacteria can easily utilize available sources of carbon and nutrients, and thus degrade organic pollutants with lower efficiency. This may also explain why, in the phytoremediation experiments inoculated with the live strain, higher degradation of PH was not observed, compared to other enhanced phytoremediation experiments. In previous bioremediation studies using *Rhodococcus erythropolis* CD106 strain isolated from bulk soil strongly contaminated with PHs, higher survival of the inoculant was observed in the case of bioaugmentation as compared to enhanced phytoremediation [20]. The number of rifampicin-resistant cells of the CD106 strain in both treatments decreased during the experiment and after 210 days of incubation reached a value of 1.4 and $3.5 \times 10^4$ cfu g$^{-1}$ d.w. of soil for enhanced phytoremediation and bioaugmentation, respectively. In contrast to the CD106 strain, CDEL254 is an endophyte and its high nutritional requirements might be the reason for improved survival in a nutrient-rich rhizosphere. The CDEL254 strain could rapidly colonize the roots of ryegrass after inoculation into the soil. Rifampicin resistant CDEL254 cells were isolated from the roots in both EP and EPRh experimental groups as early as 24 h after soil inoculation. In the first month of the experiment, their numbers temporarily increased and subsequently started to decrease. At the end of the experiment, rifampicin resistant CDEL254 cells were isolated only from the roots of plants from the EPRh treatment. The potential of the tested strain to colonize plants was also confirmed by its temporary presence in the shoots of treated plants. It is difficult to explain the reason behind such observations. The reduction in the number of inoculants on selective media could be due to bacterial death in adverse environments or their transition into a VBNC-like state. The results of study performed by Mclnroy et al. (1996) [59] on colonization of internal tissues of cotton (*Gossypium hirsutum* cv. DES-119) by seven rifampicin-resistant endophytic bacteria, demonstrated the difficulty in the exact determination of the number of rifampicin-resistant inoculants, by what is termed "antibiotic masking." This phenomenon occurs due to disturbances in the expression of genes responsible for antibiotic resistance caused by the compounds contained in the internal plant extract. Another reason for the difficulty in determining the survival of inoculants could be the plant and its associated microbiome and metabolism. Plants have their own bacterial microbiome, which inhabits the root surface, rhizosphere soil, and the internal tissues of plants (endosphere), and is directly affected by root secretions. The plant microbiome is essential for plant growth, health, and stress resilience [60]. Each plant species and cultivar usually host a specific core microbiome. The bacterial community composition in the rhizo- and endosphere changes with the development stage of a plant and in response to environmental changes. A reduction in the number of inoculants in the interior of plants can be the result of these natural changes.

*4.3. The Impact of the Applied Treatment on the Total Bacterial Number in the Soil*

An important part of bioremediation studies is monitoring the influence of the applied treatment on the number of indigenous bacteria [61]. In the current study, this issue was studied by analyzing the total number of bacteria in soil at the beginning and at the end of the experiment. We observed a significant decrease in the number of 16S rRNA copy numbers in all non-planted experimental

groups treated with rhamnolipid compared to the NA soil both on day one and 112. This negative effect was lower in the case of planted soil. These results indicate a negative impact of rhamnolipid on soil autochthonous bacteria. Similarly, Zhao et al. (2020) [62] observed an inhibitory effect of rhamnolipid on tested bacterial species (*Escherichia coli*, *Staphylococcus aureus*, and *Micrococcus aureus*). Bharali et al. (2013) [63] reported that rhamnolipids at concentrations above their critical micelle concentration (CMC) can cause bacterial cell death by destroying the phospholipid bilayer in the cell membrane. In our experiment, although rhamnolipid was applied at concentrations much below its CMC, its inhibitory effect was still observed. Contrastingly, the decrease in the 16S rRNA gene number could result from rapid hydrocarbon bioavailability caused by rhamnolipid addition. Nevertheless, the observed decrease in gene number, caused by rhamnolipid, was smaller in all soils planted with *L. perenne*, than in non-planted soils. This may indicate a protective role of plants in counteracting the adverse effect of rhamnolipid treatment. In PH-polluted soils, plants are able to stabilize, take up, accumulate, and/or transform pollutants, as well as stimulate rhizoremediation of contaminants by supporting the indigenous hydrocarbon-degrading microorganisms via release of root exudates [64,65]. In such a plant-impacted environment, the degradation of pollutants can occur at a higher rate.

## 5. Conclusions

The simultaneous use of several methods to support the bioremediation of soils highly polluted with PHs does not always produce the expected results. In the current study, the use of an *R. erythropolis* CDEL254 strain increased the fresh biomass of shoots and roots; however, the simultaneous addition of rhamnolipid inhibited this effect. Similarly, the use of Rh together with biostimulation (dead biomass) and/or bioaugmentation (living cells) does not significantly increase the efficiency of the process. This highlights the complexity of the interaction between microorganisms, plants, and pollutants. However, the use of biological methods to increase the degradation potential of the indigenous microorganisms of the soil or to enhance the bioavailability of pollutants significantly influences the rate of pollutant removal from the soil.

**Author Contributions:** Conceptualization, T.P.; methodology, T.P., M.P.-P.; software, T.P.; validation, T.P., M.P.-P. and M.N.; formal analysis, T.P., M.P.-P.; investigation, T.P., N.P., M.P.-P., M.N.; resources, T.P.; data curation, T.P.; writing—original draft preparation, T.P., M.P.-P., M.N.; writing—review and editing, T.P.; visualization, T.P., M.P.-P.; supervision, T.P.; project administration, T.P.; funding acquisition, T.P. All authors have read and agreed to the published version of the manuscript.

**Funding:** The research was supported by grant No. 2016/23/D/NZ9/01400 (SONATA 12) financed by the National Science Centre (Poland).

**Conflicts of Interest:** The authors declare no conflict of interest.

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
