# Peer review of "Comparative Study on Multiway Enhanced Bio- and Phytoremediation of Aged Petroleum-Contaminated Soil"

_agronomy, doi:10.3390/agronomy10070947_

Round 1

Reviewer 1 Report

The manuscript deals with the ‘Comparative study on multiway enhanced bio- and phyto-remediation of aged petroleum-contaminated soil’ and, it is a good and novel aspect for microbe assisted phytoremediation of aged petroleum-contaminated soils. The authors have attempted to explore the activity of Bio- and phyto-remediation of soil polluted with petroleum hydrocarbons using Rhodococcus erythropolis/Pseudomonas aeruginosa and/or ryegrass (Lolium perenne) respectively. However, the manuscript has a few minor errors that need to be addressed by the authors.

Comments

Abstract

Include Lolium perenne L. cv. Pearlgreen in abstract  

  1. Introduction

The authors need to report the recent studies of microbe-assisted phytoremediation of

 PHs.

  1. Materials and Methods

Missing source of microbes like plant names.

What was the reason to choose rifampicin-resistant mutant strains for this study?  

Size of soil volume and pots type, seed source?

Inoculum preparation in brief

No experimental data 16S rDNA PCR procedure, sequencing, and phylogenetic analysis, and tree type.

  1. Results

Rhodococcus degradans, Rhodococcus qingshengii, Rhodococcus erythropolis-italic

References

Not in a single format

Ex;

Bharali, P., Saikia, J.P., Ray, A., Konwar, B.K., 2013. Rhamnolipid (RL) from Pseudomonas aeruginosa OBP1: A novel chemotaxis and antibacterial agent. Colloids Surfaces B Biointerfaces 103, 502–509. 567 https://doi.org/10.1016/j.colsurfb.2012.10.064

Author Response

Reviewer#1

Abstract

Include Lolium perenne L. cv. Pearlgreen in abstract  

 DONE L. 26

Introduction

The authors need to report the recent studies of microbe-assisted phytoremediation of PHs

DONE  L. 89-94; A few examples of modern approaches was incorporated in the text.

Materials and Methods

Missing source of microbes like plant names.

DONE L. 109-110;

What was the reason to choose rifampicin-resistant mutant strains for this study?  

DONE L. 171-172; In the preliminary studies we did not find rifampicin-resistant strains in soil and plants used in pot experiment. The gene conferring the resistance of bacteria to rifampicin is located in the genome, what greatly reduces the risk of gene transfer to other strains. To confirm the identification, the colonies grown on rifampicin-supplemented media were additionally analysed based on their morphology and fatty acid profiles. Thus we are confident that the rifampicin-resistant bacteria isolated from the soil and plant tissues on media containing rifampicin were the same bacteria that we introduced into the soil during the experiment.

Size of soil volume and pots type, seed source?

DONE L. 211-214;

Inoculum preparation in brief

DONE L. 181-187;

No experimental data 16S rDNA PCR procedure, sequencing, and phylogenetic analysis, and tree type.

DONE L. 150-161

Results

Rhodococcus degradans, Rhodococcus qingshengii, Rhodococcus erythropolis-italic

DONE During formatting, the system changed the selected Latin names. We have checked and corrected italics for whole manuscript.

References

Not in a single format

Ex;

Bharali, P., Saikia, J.P., Ray, A., Konwar, B.K., 2013. Rhamnolipid (RL) from Pseudomonas aeruginosa OBP1: A novel chemotaxis and antibacterial agent. Colloids Surfaces B Biointerfaces 103, 502–509. 567 https://doi.org/10.1016/j.colsurfb.2012.10.064

DONE

We didn't notice problems with Mendeley software before submitting. Now all formatting errors have been corrected.

Reviewer 2 Report

Additional designed information is needed for the Methods/Experimental Design sections-:

For instance, line 111- "1-aminocyclopropane-1-carboxylic acid deaminase (ACC deaminase and cellulase activity were estimated." How were they estimated? How were isolated "examined"? How was the 1% (v/v) oil added to the plates/media? What were the conditions for the 16s rRNA amplification  (e.g., temperatures, times). 

More information is needed in most of the methods sections in order to reproduce the experiments and confirm the validity of the experimental methods.

Figure 5, the column labels are unclear (e.g., what is 10,1?). The row labels are facing the opposite directions than the previous figures.

Photos of the plants and treatment cells would be useful.

Why were the plant leaves not checked for the presence of endophytes? Why only the shoots and roots?

The discussion is a bit difficult to follow at times and could be better arranged for clarity and brevity. 

Author Response

Reviewer#2

Additional designed information is needed for the Methods/Experimental Design sections-:

For instance, line 111- "1-aminocyclopropane-1-carboxylic acid deaminase (ACC deaminase and cellulase activity were estimated." How were they estimated? How were isolated "examined"? How was the 1% (v/v) oil added to the plates/media?

DONE Missing information has been completed in Materials and Methods, especially in chapters 2.1 (L. 113-115; L. 117-139; L. 150-161) and 2.3 (L. 172-173;L. 181-187)

What were the conditions for the 16s rRNA amplification  (e.g., temperatures, times). 

DONE The condition for the 16S rRNA amplification was incorporated L.150-161.

More information is needed in most of the methods sections in order to reproduce the experiments and confirm the validity of the experimental methods.

DONE

Figure 5, the column labels are unclear (e.g., what is 10,1?). The row labels are facing the opposite directions than the previous figures.

DONE Figure 5 has been corrected

Photos of the plants and treatment cells would be useful.

Unfortunately, we do not have photographic documentation of this experiment.

Why were the plant leaves not checked for the presence of endophytes? Why only the shoots and roots?

DONE Endophytic bacteria were isolated from roots and leaves. In the case of grasses most of aboveground parts, named here as shoots, are leaves. We explained the terms in L 223-224.

The discussion is a bit difficult to follow at times and could be better arranged for clarity and brevity. 

DONE To improve the clarity we divided the discussion into chapters.

Reviewer 3 Report

Overall: This is a well written manuscript with its present form.

Abstract: Concise yet precise with all the required information that a good abstract should include in itself.

Introduction: It is nicely written providing the reader with background of bioremediation and phytoremediation of organic compounds and knowledge gaps which need to be fulfilled highlighting the need of this research study.

Materials and Methods: The experimental design and all the analytical and statistical methods are very well explained in a detailed manner.

Results: The results of the study are well explained and highlighted with statistical significance.

Discussion: This section is pretty detailed as well and research outcomes from this study are also compared with research done by previous authors.

Conclusions: The authors do not overstate the significance of their results and recognize the shortcomings in their research findings which would encourage further research in this field to fill in the knowledge gaps.

Author Response

Reviewer#3

According to Reviewer#3 revision was not required

Abstract: Concise yet precise with all the required information that a good abstract should include in itself.

Introduction: It is nicely written providing the reader with background of bioremediation and phytoremediation of organic compounds and knowledge gaps which need to be fulfilled highlighting the need of this research study.

Materials and Methods: The experimental design and all the analytical and statistical methods are very well explained in a detailed manner.

Results: The results of the study are well explained and highlighted with statistical significance.

Discussion: This section is pretty detailed as well and research outcomes from this study are also compared with research done by previous authors.

Conclusions: The authors do not overstate the significance of their results and recognize the shortcomings in their research findings which would encourage further research in this field to fill in the knowledge gaps.